# Prediction Models and Decision Aids for Women with Ductal Carcinoma In Situ: A Systematic Literature Review

**DOI:** 10.3390/cancers14133259

**Published:** 2022-07-02

**Authors:** Renée S. J. M. Schmitz, Erica A. Wilthagen, Frederieke van Duijnhoven, Marja van Oirsouw, Ellen Verschuur, Thomas Lynch, Rinaa S. Punglia, E. Shelley Hwang, Jelle Wesseling, Marjanka K. Schmidt, Eveline M. A. Bleiker, Ellen G. Engelhardt, Grand Challenge PRECISION Consortium

**Affiliations:** 1Department of Molecular Pathology, Netherlands Cancer Institute, 1066 CX Amsterdam, The Netherlands; r.schmitz@nki.nl (R.S.J.M.S.); j.wesseling@nki.nl (J.W.); mk.schmidt@nki.nl (M.K.S.); 2Department of Scientific Information Service, Netherlands Cancer Institute, 1066 CX Amsterdam, The Netherlands; e.wilthagen@nki.nl; 3Department of Surgery, Netherlands Cancer Institute, 1066 CX Amsterdam, The Netherlands; f.v.duijnhoven@nki.nl; 4Borstkanker Vereniging Nederland, 3511 DT Utrecht, The Netherlands; oirsouw1958@hetnet.nl (M.v.O.); eajverschuur@gmail.com (E.V.); 5Division of Surgical Oncology, Duke University, Durham, NC 27708, USA; thomas.lynch2@duke.edu (T.L.); shelley.hwang@duke.edu (E.S.H.); 6Department of Radiation Oncology, Dana-Farber Cancer Institute, Boston, MA 02215, USA; rpunglia@partners.org; 7Department of Pathology, Leiden University Medical Center, 2333 ZA Leiden, The Netherlands; 8Department of Pathology, Nethelands Cancer Institute, 1066 CX Amsterdam, The Netherlands; 9Department of Psycho-Oncology and Epidemiology, Netherlands Cancer Institute, 1066 CX Amsterdam, The Netherlands; e.engelhardt@nki.nl

**Keywords:** ductal carcinoma in situ, decision support tool, decision aid, prediction model

## Abstract

**Simple Summary:**

Ductal carcinoma in situ (DCIS) is a potential precursor to invasive breast cancer (IBC). Although in many women DCIS will never become breast cancer, almost all women diagnosed with DCIS undergo surgery with/without radiotherapy. Several studies are ongoing to de-escalate treatment for DCIS. Multiple decision support tools have been developed to aid women with DCIS in selecting the best treatment option for their specific goals. The aim of this study was to identify these decision support tools and evaluate their quality and clinical utility. Thirty-three studies were reviewed, in which four decision aids and six prediction models were described. While some of these models might be promising, most lacked important qualities such as tools to help women discuss their options or good quality validation studies. Therefore, the need for good quality, well validated decision support tools remains unmet.

**Abstract:**

Even though Ductal Carcinoma in Situ (DCIS) can potentially be an invasive breast cancer (IBC) precursor, most DCIS lesions never will progress to IBC if left untreated. Because we cannot predict yet which DCIS lesions will and which will not progress, almost all women with DCIS are treated by breast-conserving surgery +/− radiotherapy, or even mastectomy. As a consequence, many women with non-progressive DCIS carry the burden of intensive treatment without any benefit. Multiple decision support tools have been developed to optimize DCIS management, aiming to find the balance between over- and undertreatment. In this systematic review, we evaluated the quality and added value of such tools. A systematic literature search was performed in Medline(ovid), Embase(ovid), Scopus and TRIP. Following the PRISMA guidelines, publications were selected. The CHARMS (prediction models) or IPDAS (decision aids) checklist were used to evaluate the tools’ methodological quality. Thirty-three publications describing four decision aids and six prediction models were included. The decision aids met at least 50% of the IPDAS criteria. However, most lacked tools to facilitate discussion of the information with healthcare providers. Five prediction models quantify the risk of an ipsilateral breast event after a primary DCIS, one estimates the risk of contralateral breast cancer, and none included active surveillance. Good quality and external validations were lacking for all prediction models. There remains an unmet clinical need for well-validated, good-quality DCIS risk prediction models and decision aids in which active surveillance is included as a management option for low-risk DCIS.

## 1. Introduction

Ductal carcinoma in situ (DCIS) is a potential precursor lesion to invasive breast cancer (IBC). It accounts for approximately 20% of all newly screen-detected breast lesions [1,2]. It is mostly non-symptomatic, and detected through population-based screening. As DCIS is historically considered a potential precursor lesion for IBC, it is usually treated with breast-conserving surgery, often supplemented with radiotherapy, or even mastectomy. In several countries, endocrine treatment may also be prescribed. However, many DCIS lesions will never progress to IBC during the patient’s lifetime [3]. Biopsy review studies of patients where DCIS was initially misdiagnosed as benign and thus not treated after biopsy suggest that up to 85% of all DCIS will never progress into IBC [3,4,5,6]. Consequently, there is a growing concern about possible overtreatment for low-risk, with favorable characteristics, DCIS [3,7,8,9]. DCIS treatment de-escalation trials are being conducted to investigate the safety of active surveillance in women with low-risk DCIS [10,11,12].

Decision making about DCIS treatment is not straightforward. The key difference is the impact these treatments can have on patients’ quality of life [13]. Evidence on patient reported outcomes in women with DCIS is still lacking [14]. Active surveillance may potentially safeguard patients’ quality of life by allowing them to forego the invasive standard treatments, thereby avoiding potential harm (e.g., psychological stress and side effects and complications of surgery, radiotherapy and endocrine treatment) [15]. By not removing the DCIS lesion, however, there is a risk of progression to IBC; this knowledge might cause some women to experience elevated levels of anxiety. Patients (in consultation with their doctors) need to evaluate the risks and benefits of both options, and decide what is of most importance to them. lowering their risk of experiencing invasive breast cancer or avoiding potentially unnecessary treatments and their associated harms. This is a preference-sensitive decision that needs to be based on individual patients’ informed preferences.

Currently, women with DCIS and their clinicians already have a choice in type of surgery (i.e., breast conserving surgery or mastectomy) and the use of adjuvant treatment (i.e., radiotherapy and/or endocrine treatment). However, in the future, and also currently in the context of the ongoing DCIS treatment de-escalation trials, women with low-risk DCIS may also have the added alternative of active surveillance as a potential management strategy. Clinicians will therefore need tools to inform women with low-risk DCIS about the risks and benefits of active surveillance as a potential management strategy [16]. Patients need to receive clear and balanced information to enable them to make an informed decision.

Risk perception and views on the role of active surveillance in the management of DCIS differ among clinicians [17,18]. In addition, several studies have described overestimation of risk of recurrence/progression and lack of knowledge about available management strategies and their associated risks, both physical and psychological, amongst women diagnosed with DCIS [13,15,19,20]. Women searching for information may turn to the internet to meet their needs. However, Blackwood et al. reported that few of the plain language summaries available on the internet met quality criteria for consumer health information [21], thus highlighting the need for properly designed DCIS decision support tools.

Well-designed evidence-based decision support tools such as decision aids, communication tools or prediction models may help patients and clinicians make better informed and value-congruent decisions. Decision aids – tools developed to support patients facing preference-sensitive decisions – provide balanced and easily accessible plain language information about all viable options; they may contain value clarification exercises or question prompt lists to help patients discuss their options with medical professionals [22]. Patients who have used decision aids often report feeling better informed about DCIS; this may result in more accurate risk perception [23]. Communication aids are tools developed to help clinicians provide their patients with balanced, easy-to-understand information during the clinical encounter [24]; this may serve to increase effective patient participation in decision making. Finally, prediction models, statistical models used to quantify individualized risks of experiencing specific outcomes (e.g., a recurrence) during a certain period, could also help clinicians and patients to evaluate the risks and benefits of available treatment options. Women with low-risk DCIS, as well as healthcare professionals involved in their care, may therefore benefit from having access to decision support tools when deciding whether to undergo surgery or opt for active surveillance (if/when available). The aim of this study is to identify and evaluate the methodological quality of published decision support tools, developed to support decision making about the management of DCIS.

## 2. Methods

### 2.1. Literature Search

A systematic literature review was performed following the Preferred Reporting Items for Systematic Reviews and Meta-Analysis (PRISMA) statement [25,26]. The protocol was published in the international prospective register of systematic reviews [27,28] (PROSPERO, CRD42020212297). The systematic search was designed and executed in collaboration with a medical information specialist (EAW). The databases Medline [29] (ovid), Embase [30] (ovid), Scopus [31] and TRIP [32] were initially searched up to 24 September 2020 (inclusive). A search update was performed on 22 February 2022. The following terms, including synonyms and closely related words, were used as index terms or free-text words: “DCIS” and “decision support systems” (Appendix A provides the full search strategy). No limits were applied for date or study design.

### 2.2. Selection Criteria

For this study, the selection criteria were: (1) studies describing the development and/or evaluation of a decision support tool (e.g., a patient decision aid, communication tool or prediction model) aimed at women with DCIS; and (2) the article was written in English or Dutch. Decision aids aimed at women with invasive breast cancer were evaluated in a recently published systematic review by Vromans et al. [33]. Thus, decision aids designed for women with invasive breast cancer with a secondary focus on DCIS were not included in the current study.

### 2.3. Literature Screening

Papers were imported and duplicates were removed. The unique papers were screened based on title and abstract by two authors (RSJMS and EGE) independently using Rayyan QRCI [34,35]. Full-text versions were retrieved for the papers selected based on title and abstract. All full texts were screened by two authors independently (RSJMS and EGE), and disagreements were resolved through consensus. A cross-reference check and a search for papers cited by or citing the publications were selected based on the full-text performed in Scopus and Web of Science and this process was repeated until there were no more new relevant papers found (February 2021). Article selection was also performed by two researchers (RSJMS and EGE) independently, with any disagreement resolved through consensus.

### 2.4. Quality Assessment and Data Extraction

The (methodological) quality of the decision aids themselves and papers describing the development and/or evaluation of the decision aids were assessed using the IPDAS (International Patient Decision Aid Standards) checklist, which reflects the gold standard in the field of decision aid development [36]. The IPDAS checklist consists of 74 items distributed over three key domains: content (30 items); development process (36 items); and effectiveness (8 items) of the decision aid. The criteria were scored as either ‘met’ or ‘unmet’. For each decision aid, we calculated the percentage of criteria that had been met per IPDAS domain. A test set of three DA were scored by both reviewers (RSJMS and EGE) independently, and disagreement was resolved through consensus. Thereafter, the remaining decision aids were scored by one reviewer (RSJMS), as there was 97% congruence in scoring.

The methodological quality of the development and validation of the DCIS prediction models was assessed using the widely endorsed CHARMS (Critical Appraisal and Data Extraction for Systematic Reviews of Prediction Modelling Studies) checklist [37,38]. The CHARMS-PF checklist (items to extract from the primary studies [38]) consists of 32 criteria spread across nine domains (Source of Data; Participants; Outcomes; Prognostic Factors; Sample Size; Missing Data; Analysis; Results; and Interpretation and Discussion). For presentation purposes, we grouped the nine CHARMS methodological quality domains into three overarching categories, namely: Participants (consisting of the CHARMS domains Source of Data, Participants and Sample Size), Methodology (consisting of the CHARMS domains Outcomes, Prognostic Factors, Missing Data and Analysis) and Results (consisting of the CHARMS domains Results and Interpretation and Discussion). For each overarching category, risk of bias is presented as low, moderate or high. All papers were double-coded, and thus individually scored by both reviewers (RSJMS and EGE), and disagreement was resolved through consensus.

## 3. Results

Of 12,000 papers screened, 33 were included in the final review (Flowchart, Figure 1). These papers described three decision aids, a communication aid and six prediction models.

### 3.1. Decision Aids and Communication Tool

Three decision aids were retrieved (Table 1), specifically: a decision aid for the German context by Berger-Hoger et al. [39,40,41]; OnlineDeCISion.org by Ozanne et al. [42,43]; DCISoptions.org by the COMET research team [44]; and one communication aid by De Morgan et al. [24]. Two of the four decision aids were aimed solely at women with DCIS [41,44], whilst onlineDeCISion.org provides a separate interface for patients and clinicians. The communication aid by De Morgan et al. was aimed at clinicians, and intended for use during consultations with the patient. The onlineDeCISion.org tool contains a disease simulation model integrating data from the published literature to simulate clinical events [45]. Three of the tools provided information in English. The decision aid by Berger-Hoger et al. was developed for German patients, and therefore written in German.

For three decision aids, both women with DCIS and healthcare professionals were included in the development process [24,39,42], while for DCISoptions.org, this was not reported. Both Berger-Hoger et al. and De Morgan et al. performed an evaluation study for their respective tools, which showed that using the tool stimulated patient involvement in decision making. The decision aid by Berger-Hoger et al. achieved the highest score for methodological quality, but all decision aids met at least 50% of the IPDAS criteria (Appendix A). Areas requiring improvement were: presentation of probability in more than one format (words, diagrams, etc.); inclusion of methods for clarifying and expressing patient’s values; guidance in deliberation and communication; and providing references to the evidence on which the decision aid was based.

### 3.2. Prediction Models

Six prediction models were described in 27 papers (Table 2). These include four classical prediction models using clinicopathological factors–two of which were designed to predict the risk of ipsilateral invasive breast cancer (iIBC) and one to predict the risk of contralateral breast cancer (CBC)–and two models including biomolecular factors.

### 3.3. iIBC Models

Three classical prediction models, predicting the risk of iIBC after DCIS, were the first models to be developed for DCIS: the Van Nuys Prognostic Index (VNPI) [46,47,48,49,50,51,52,53,54,55,56,57]; Memorial Sloane Kettering Cancer Centre Nomogram (MSKCC nomogram) [58,59,60,61]; and the DCIS patient prognostic score [62]. The VNPI and MSKCC nomogram were developed on small (n= 238 to 202 patients), single-centre cohorts of patients diagnosed between 1979 and 2006, and treated with surgery with or without radiotherapy and endocrine treatment. The DCIS patient prognostic score was developed using 32,144 patients from the SEER database diagnosed between 1988 and 2007 and only included patients treated with BCS with or without radiotherapy. Measures widely recommended by experts for assessing model performance (e.g., C-index for discriminatory accuracy and calibration) were frequently not reported. For example, no C-index was reported in the studies evaluating the VNPI [46,47,48,49,50,51,52,53,54,55,56,57] and DCIS patient prognostic score [62]. The measures used to report discriminatory accuracy, e.g., Kaplan Meier curves and descriptive statistics, are suboptimal. For the MSKCC nomogram, calibration was reported to be imperfect to good, and the C-index varied from 0.61 in a validation study [61] to 0.69 in the development study [58]. The CHARMS risk of bias was high/moderate for the VNPI and moderate for the DCIS prognostic score and the MSKCC nomogram.

### 3.4. CBC Model

One model predicting risk of developing contralateral breast cancer after DCIS was retrieved, the CBC risk model [63], which was developed using the SEER database. The CBC risk model was developed for patients with all grades of DCIS, IBC, or a combination of DCIS/IBC. The type of treatment received and follow-up duration for the development set were not reported. C-index and calibration were also not reported. Furthermore, no external validation was reported. Risk of bias according to the CHARMS checklist was moderate.

### 3.5. Biomolecular Models

Two prediction models containing biomolecular features (such as immunohistochemistry markers and gene expression) were retrieved: Oncotype DCIS [64,65,66,67] (derived from Oncotype DX [68], developed for women with invasive breast cancer) and DCISionRT [69,70]. Oncotype DCIS was developed in a trial cohort of 327 patients. DCISionRT was developed in a multicentre cohort of 526 patients. Patients in both studies were treated with BCS with or without radiotherapy and endocrine treatment (DCISionRT). Diagnosis years varied from 1986 to 2004. The number of events was very low (*n* = 46) in the development study of Oncotype DCIS, and not reported for DCISionRT. Area under the curve (AUC), C-index and calibration were not reported for DCISionRT; for Oncotype DCIS, calibration was reported to be good, and AUC was 0.68 according to a validation study by Paszat et al. [67]. Both models were developed in highly selected and relatively small patient samples, with a modest number of events (*N* < 550). Consequently, the CHARMS risk of bias was moderate for both models.

### 3.6. Clinical Utility

For all six prediction models, clinical utility remains unclear due to the highly selective development and validation datasets used, and due to limited good quality validation studies. The number of validation studies varied from zero to nine. For the VNPI [46], for example, ten validation studies were published. However, most were performed in small, highly selected patient samples consisting partly of the development sample, with low numbers of events (range 11–165) (Table 2); only Kaplan Meier curves and descriptive statistics were used to report on the discriminatory value of the model. Thus, it is very difficult to evaluate clinical utility. Similarly, Oncotype DCIS [64] was validated in three separate papers, all using the same Ontario DCIS cohort [71]. For the DCIS RT score [62] and the CBC risk model [63], no validation studies were retrieved.

## 4. Discussion

By performing a systematic literature review, we inventoried and assessed the quality of available DCIS decision support tools. We retrieved three decision aids, one communication aid and six prediction models. All decision aids included the option of active surveillance; the communication aid by de Morgan et al. did not. There is room for improvement amongst all the decision aids, but they may serve as templates for the development of other novel aids for informing patients about their treatment options for DCIS. There are few decision aids available, but most are only available in English. None of the six published prediction models included the option of active surveillance, thus there is a need to extend existing models to include this option or develop new tools. Furthermore, assessment of and reporting on the performance of the models was generally suboptimal. The clinical utility of the available models will remain unclear until additional, good-quality external validations are performed in adequately sized cohorts with sufficient events.

Patient decision aids have been shown to effectively provide patients with balanced information on all available treatment strategies, and help them make value-congruent decisions [23]. Decision aids on surgery for women with early-stage breast cancer, for example, have been shown to improve patient involvement, patient knowledge and decision-related outcomes such as decisional conflict, satisfaction and overall quality of life [72]. Women with low-risk DCIS may face similar treatment decisions to women with early-stage breast cancer. However, they have a different prognosis and may in the future have the option to forego surgery and opt for active surveillance. Therefore, well-designed (according to the IPDAS criteria [36]) and properly evaluated decision aids, including the benefits and harms associated with active surveillance, need to be developed to support women with (low-risk) DCIS.

The literature shows that women diagnosed with DCIS have clear knowledge gaps. These misconceptions, particularly incorrect estimation of recurrence/progression risk, can cause increased worry and anxiety and impact negatively upon quality of life [19,73,74,75].

Prediction models and decision aids for DCIS are not widely used in daily practice. As a result, their applicability and impact are not well studied. For the prediction models reviewed, validation studies were limited in number and/or quality. Similarly, for the decision aids, no implementation studies were performed. A lack of validation and implementation studies might therefore hinder the decision support tools in being fully implemented within the clinic setting.

The limited availability of large, population-based datasets of women diagnosed with DCIS that include sufficient follow-up time and events is a barrier for the development of good prediction models. Consequently, many of the models we identified were developed and validated using small, highly selected populations (e.g., trial data and single centre cohorts) with limited follow-up and a small number of events. Thus, the clinical utility of these models for the general DCIS population remains unclear. For example, in a US-based general community cohort of 91 DCIS patients, there was limited agreement in local breast event risk estimates when comparing the VNPI, MSKCC nomogram, Oncotype DCIS and risk estimates from three radiation oncologists [76]. The predicted risks were so highly divergent that it is difficult to determine what should be the “gold standard”. This large divergence in risk estimates is not seen in all studies; Van Zee et al., for example, have reported a 92% concordance in the estimates of loco-regional recurrence risk generated by the MSKCC nomogram and Oncotype DCIS in a dataset with 59 US-based women with DCIS [77].

These examples stress the need for new prediction models to be designed specifically for women with DCIS, but also the need for rich datasets with information for large representative cohorts. Previous publications have shown that translating prediction models that focus on invasive breast cancer to DCIS might be challenging. For instance, a model designed to predict risk of developing contralateral breast cancer for women treated for invasive breast cancer (PredictCBC [78]) was applied to a cohort of Dutch DCIS patients [79]. This model did not perform well in the DCIS cohort, and one of the reasons for this was that many of the strong predictors in the model were not available in the dataset. In this model, hormone receptor status and BRCA status were important predictors, but as these variables are not routinely collected for women with DCIS, the model could not be applied optimally. Similarly, Oncotype DCIS was derived from Oncotype DX, the prediction model developed for women with invasive breast cancer, which had to be extensively adapted to apply to women with DCIS [64]. Thus, models that predict risk of developing invasive breast cancer for women without a history of breast cancer, such as the Breast Cancer Risk Assessment Tool (BCRAT) or Breast and Ovarian Analysis of Disease Incidence and Carrier Estimation Algorithm (BOADICEA), cannot simply be applied to a woman with a history of DCIS [80,81].

The prediction models described in our study used similar clinicopathological factors (such as size, DCIS grade and margin width). Model performance was rarely reported, but when available, most models showed modest performance at best. We also included more recently developed models containing biomolecular features and arrays. However, due to limited validation and reporting of performance measures, it is currently unclear if these models perform adequately for women with DCIS. A promising next step towards achieving improved models is moving into the realm of artificial intelligence and machine learning. Such techniques could provide interesting novel options to explore in the context of DCIS. For example, Klimov et al., developed a novel machine learning pipeline to predict risk of ipsilateral breast cancer after DCIS using digitized whole slide images and clinicopathologic long-term outcome data [82], thereby offering a promising new direction.

None of the prediction models we retrieved included the option of active surveillance. As clinical trials studying the safety of active surveillance for women with DCIS are still ongoing, and thus active surveillance is not yet a standard management strategy for DCIS, adequately predicting the outcome of active surveillance is challenging, as it is not yet offered regularly to women with low-risk DCIS. A promising new decision support tool combining a risk calculator with a decision aid that included almost 2000 patients receiving active surveillance in the United States was recently published by Fridman et al. [83]. However, the development paper of the risk calculator within this decision support tool has not yet been published.

A limitation of our study is that we used a search of scientific publications to identify decision support tools. We did not perform an extensive search for unpublished web-based decision support tools for women with DCIS. However, Blackwood et al. performed an extensive search to identify internet-based information resources for DCIS in which they retrieved mostly plain language summaries or informational websites. Only two decision aids were retrieved, neither of which was specifically for women with DCIS [21]. Therefore, we expect that it is unlikely that we have omitted unpublished DCIS decision aids available online.

## 5. Conclusions

To our knowledge, our systematic review is the first to provide an extensive overview of the available decision support tools for women with DCIS. From our study, we can conclude that there are only a few decision support tools available for women with DCIS, and these tools are mainly in English. The available decision aids are promising, but they do require improvement (e.g., addition of components to facilitate communication with healthcare professionals) to maximize their usefulness and clinical utility, and there is a need for clinical evaluation studies to establish their effectiveness. Based on the available evidence, none of the prediction models retrieved are ready to be implemented in daily clinical practice for women with DCIS. Additional validation studies in larger, more diverse populations are urgently needed to establish the clinical utility of these models. Furthermore, prediction models must be extended, or new models developed, to include the option of active surveillance for women diagnosed with low-risk DCIS to align this area of research with ongoing clinical developments.

## Figures and Tables

**Figure 1 cancers-14-03259-f001:**
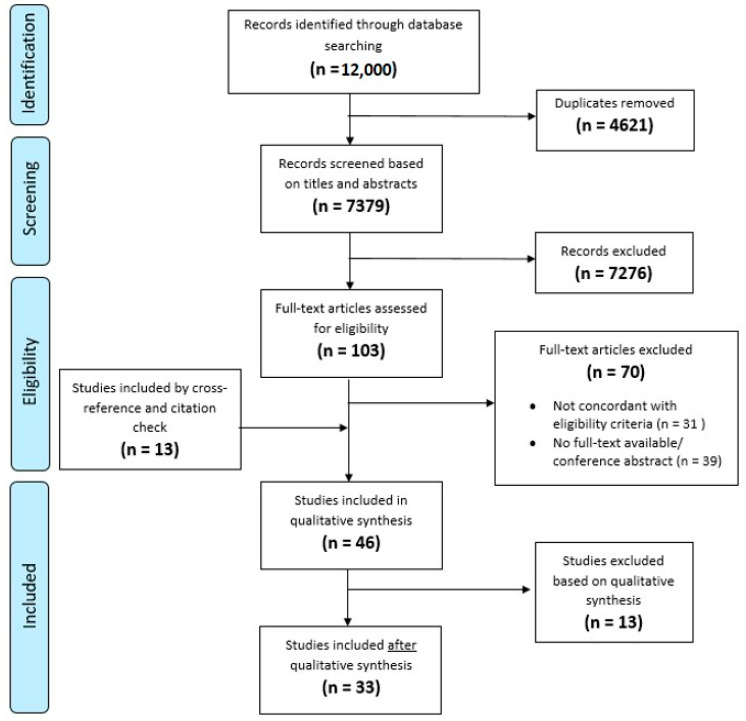
Flowchart inclusion of papers.

**Table 1 cancers-14-03259-t001:** Overview of patient decision aids on treatment decision for (low-risk) DCIS.

	Decision Aid Berger-Hoger et al.	Communication Aid De Morgan et al.	Online DeCISion.org Ozanne et al.	DCISoptions.org COMET Trial Decision Aid
Last update	Not reported	Not reported	Not reported	Not reported
Language	German	English	English	English
Country	Germany	Australia	USA	USA
Format	On paper only	Online booklet *	Web-based **	Web-based ***
Target audience	Women with DCIS	Healthcare professionals	Healthcare professionals and women with DCIS	Women with DCIS
Patients involved in development	Yes	Yes	Yes	Not reported
Healthcare professionals involved in development	Yes	Yes	Yes	Not reported
Evaluation study conducted	Yes	Yes	Not reported	Not reported
Design evaluation study	RCT	Interview	N.A.	N.A.
Sample size evaluation study	64	25	N.A.	N.A.
Main finding evaluation study	More patient involvement	Communication tool assists SDM	N.A.	N.A.
Implementation study conducted	Not reported	Not reported	Not reported	Not reported
IPDAS score for CONTENT (% criteria met)	87%	57%	65%	78%
IPDAS score for DEVELOPMENT (% criteria met)	71%	59%	67%	42%
IPDAS score for EFFECTIVENESS (% criteria met)	100%	50%	75%	75%

Abbreviations: DCIS: Ductal Carcinoma in Situ, SDM: Shared decision making, IPDAS: Inter-national Patient Decision Aids Standard.

**Table 2 cancers-14-03259-t002:** Overview of prediction models predicting subsequent breast events after DCIS.

	Oncotype DCIS (Solin et al., (2013))	DCISionRT PreludeDX (Bremer et al., (2018))	Van Nuys Prognostic Index (Silverstein et al., (1995))	MSKCC DCIS Nomogram (Rudlof et al., (2010))	Patient Prognostic Score (Sagara et al., (2016))	CBC Risk Model (Chowdhury et al., (2017))
Country	USA	Sweden	USA	USA	USA	USA
Format	On order *	On order **	On paper	Web based ***	On paper	On paper
Predicted outcome	Ipsilateral in situ or invasive breast event	Ipsilateral in situ or invasive breast event	Disease-free survival	Ipsilateral in situ or invasive breast event	Breast cancer-specific death	Contralateral invasive breast cancer
Tool based on	Multigene assay	Clinicopathological factors + biomarkers	Clinicopathological factors	Clinicopathological factors	Clinicopathological factors	Clinicopathological factors
Type of data	Trial cohort	Multi center	Single center	Trial cohort	Population-based	Population-based
Number of patients	327	526	238	1868	32,144	7684
Number of events	46	Not reported	31	202	304	1921
Intended to support decision making about:	Adjuvant radiotherapy	Adjuvant radiotherapy	Type of surgery and adjuvant radiotherapy	Adjuvant radiotherapy	Adjuvant radiotherapy	Screening or prophylactic mastectomy
Risk of bias based on CHARMS	Moderate	Moderate	Moderate/High	Moderate	Moderate	Moderate
Number of validation studies retrieved	3	2	10	3	0	0
Type of data validation studies	Trial and population-based	Trial and Single center	Single- and Multi center	Single center	N.A.	N.A.
Number of patients validation studies (range)	718–1102	455–504	159–949	467–734	N.A.	N.A.
Number of events validation studies (range)	65–100	54–90	11–165	42–63	N.A.	N.A.
C-index/AUC	0.68	None reported	None reported	0.61–0.68	None reported	None reported
Clinical utility	Unclear	Unclear	Unclear	Unclear	Unclear	Unclear

Abbreviations: DCIS: Ductal Carcinoma In Situ, CBC: Contralateral Breast Cancer, DFS: Disease Free Survival, CHARMS: Critical Appraisal and Data Extraction form Systematic Reviews of Prediction Modelling Studies, AUC: Area Under the Curve.

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
