# Peer review of "Prediction Models and Decision Aids for Women with Ductal Carcinoma In Situ: A Systematic Literature Review"

_cancers, 2022, doi:10.3390/cancers14133259_

Round 1

Reviewer 1 Report

This review paper by Bleiker and colleagues is a well-researched addition to the field of understanding prediction models in breast cancer diagnosis outcomes. The authors present a strong case for metrics that need to be consistent across different prediction models, as well as a stringent review process that will make it easier for readers and diagnosticians. There are only a few minor typographical errors that I would recommend corrections on:

The formatting of the abstract doesn't need the sub-sections. 

Stray - mark on line 200

Reformat Table 2 for consistent fitting of words within the column width

Author Response

Dear Reviewers,

On behalf of the entire author team, we thank you for taking the time to review our paper. We have read the compliments and feedback with great pleasure. Below we specify how we have addressed the feedback per item.

  1. The formatting of the abstract doesn't need the sub-sections. 
    The subsections have been removed as suggested.

  2. Stray - mark on line 200
    The subtext of Table 2 was compressed. The issue has been corrected.

  3. Reformat Table 2 for consistent fitting of words within the column width
    We have adjusted and/or shortened some entries, which improved the table’s readability.
  4. There is the newest version of PRISMA Statement from 2020 so authors should check if this systematic review follows it: https://doi.org/10.1136/bmj.n71
    Thank you for bringing this to our attention. We have checked all new criteria and our review follows the new guideline. The reference has been updated.

  5. Line 43: There is no need to put quote marks.
    The quotation marks were put there because we believe it is not correct to class DCIS as breast cancer. We have adapted the term cancers to lesions to leave out the quotation marks.

  6. Line 110: Cite reference for PROSPERO register, e.g., https://doi.org/10.1016/S0140-6736(10)60903-8
    The reference has been added.

  7. For the sake of unambiguity and faster access to further information described in this manuscript, web addresses should be provided for all resources available on-line (registers, databases, guidelines, etc.).
    URLs for all databases, PROSPERO and Rayyan were added to the reference list. URLs to web-based tools were added to the subtext of each table.

  1. In Table 1 and Table 2 it should be mentioned in which format(s) are decision aids, communication tools and prediction models available (e.g., web-based tools (provide addresses), standalone software, etc.).
    Format and links to the web-addresses were added to both Table 1 and Table 2.

Again thank you for considering our paper. We hope we have addressed the feedback satisfactorily.

Kind regards,

Renée Schmitz

Reviewer 2 Report

In the submitted manuscript Schmitz et al. provided a short systematic literature review on prediction models and decision aids for women with ductal carcinoma in situ.

This manuscript is well written and covers most important publication on its topic. However, there are few thing which have to addressed be to further improve quality of this manuscript:

1) There is the newest version of PRISMA Statement from 2020 so authors should check if this systematic review follows it: https://doi.org/10.1136/bmj.n71

2) Line 43: There is no need to put quote marks.

3) Line 110: Cite reference for PROSPERO register, e.g., https://doi.org/10.1016/S0140-6736(10)60903-8

4) For the sake of unambiguity and faster access to further information described in this manuscript, web addresses should be provided for all resources available on-line (registers, databases, guidelines, etc.).

5) In Table 1 and Table 2 it should be mentioned in which format(s) are decision aids, communication tools and prediction models available (e.g., web-based tools (provide addresses), standalone software, etc.).

Author Response

(The authors gave the same response as above.)
